# Learning Representations and Generative Models for 3D Point Clouds

**Panos Achlioptas** *
Department of Computer Science
Stanford University, USA

**Olga Diamanti**
Department of Computer Science
Stanford University, USA

**Ioannis Mitliagkas**
Department of Computer Science and Operations Research
University of Montréal, Canada

**Leonidas Guibas**
Department of Computer Science
Stanford University, USA

## Abstract

Three-dimensional geometric data offer an excellent domain for studying representation learning and generative modeling. In this paper, we look at geometric data represented as point clouds. We introduce a deep autoencoder (AE) network with state-of-the-art reconstruction quality and generalization ability. The learned representations outperform existing methods on 3D recognition tasks and enable basic shape editing via simple algebraic manipulations, such as semantic part editing, shape analogies and shape interpolation. We perform a thorough study of different generative models including: GANs operating on the raw point clouds, significantly improved GANs trained in the fixed latent space of our AEs and, Gaussian mixture models (GMM). For our quantitative evaluation we propose measures of sample fidelity and diversity based on matchings between sets of point clouds. Interestingly, our careful evaluation of generalization, fidelity and diversity reveals that GMMs trained in the latent space of our AEs produce the best results.

## 1 Introduction

Three-dimensional (3D) representations of real-life objects are a core tool for vision, robotics, medicine, augmented and virtual reality applications. Recent encodings like view-based projections, volumetric grids and graphs, complement more traditional shape representations such as 3D meshes, level set functions, curve-based CAD models and constructive solid geometry (Botsch et al., 2010). These encodings, while effective in their respective domains (e.g. acquisition or rendering), are often poor in semantics. For example, naïvely interpolating between two different cars in a view-based representation does not yield a representation of an "intermediate" car. Furthermore, these raw, high-dimensional representations are typically not well suited for the design of generative models via classic statistical methods. As such, editing and designing new objects with such representations frequently involves the construction and manipulation of complex, object-specific parametric models that link the semantics to the representation. This may require significant expertise and effort.

Recent advances in deep learning bring the promise of a *data-driven approach*. In domains where data is plentiful, deep learning tools have eliminated the need for hand-crafting features and models. Deep learning architectures like autoencoders (AEs) (Rumelhart et al., 1988; Kingma & Welling, 2013) and Generative Adversarial Networks (GANs) (Goodfellow et al., 2014; Radford et al., 2015; Denton et al., 2015; Che et al., 2016) are successful at learning complex data representations and generating realistic samples from complex underlying distributions. Recently, deep learning architectures for view-based projections (Su et al., 2015; Wei et al., 2016; Kalogerakis et al., 2016), volumetric grids (Qi et al., 2016b; Wu et al., 2015; Hegde & Zadeh, 2016) and graphs (Bruna et al., 2013; Henaff et al., 2015; Defferrard et al., 2016; Yi et al., 2016b) have appeared in the 3D machine learning literature.

In this paper we focus on point clouds, a relatively unexplored 3D modality. Point clouds provide a homogeneous, expressive and compact representation of surface geometry, easily amenable to

---

*Correspondence to: Panos Achlioptas <optas@cs.stanford.edu>.

geometric operations. These properties make them attractive from a learning point of view. In addition, they come up as the output of common range-scanning acquisition pipelines used in devices like the Kinect and iPhone's recent face identification feature. Only a handful of deep architectures for 3D point clouds exist in the literature: PointNet (Qi et al., 2016a; 2017) successfully tackled classification and segmentation tasks; Kalogerakis et al. (2016) used point-clouds as an intermediate step in their pipeline; Fan et al. (2016) used pointclouds as the underlying representation to extract 3D information from 2D images. We provide the first results that use deep architectures with the focus of learning representations and generative models for point clouds.

Generative models have garnered increased attention recently in the deep learning community with the introduction of GANs (Goodfellow et al., 2014). An issue with GAN-based generative pipelines is that training them is notoriously hard and unstable (Salimans et al., 2016). More importantly, *there is no universally accepted way to evaluate generative models*. In evaluating generative models one is interested in both *fidelity*, i.e. how much the generated points resemble the actual data, and *coverage*, i.e. what fraction of the data distribution a generated sample represents. The latter is especially important given the tendency of certain GANs to exhibit mode collapse. We provide simple methods to deal with both issues (training and evaluation) in our target domain. Our specific contributions are:

- We design a new AE architecture—inspired by recent architectures used for classification (Qi et al., 2016a)—that is capable of learning compact representations of point clouds with excellent reconstruction quality even on unseen samples. The learned representations are (i) good for classification via simple methods (SVM), improving on the state of the art (Wu et al., 2016); (ii) suitable for meaningful interpolations and semantic operations.

- We create the first set of generative models which (i) can generate point clouds measurably similar to the training data and held-out test data; (ii) provide good coverage of the training and test dataset. We argue that jointly learning the representation and training the GAN is unnecessary for our modality. We propose a workflow that first learns a representation by training an AE with a compact bottleneck layer, then trains a plain GAN in that fixed latent representation. Intuitively, training a GAN inside a compact, low-dimensional representation is easier. We point to theory (Arjovsky & Bottou, 2017) that supports this idea, and verify it empirically. Latent GANs are *much easier* to train than monolithic (raw) GANs and achieve superior reconstruction with much better coverage. Somewhat surprisingly, GMMs trained in the latent space of fixed AEs achieve the best performance across the board.

- We show that multi-class GANs work almost on par with dedicated GANs trained per-object-category, as long as they are trained in the latent space.

- To support our qualitative evaluation, we perform a careful study of various old and new metrics, in terms of their applicability (i) as objectives for learning good representations; (ii) for the evaluation of generated samples. We find that a commonly used point cloud metric, Chamfer distance, fails to discriminate certain pathological cases from good examples. We also propose fidelity and coverage metrics for our generative models, based on an optimal matching between two different samples, e.g. a set of point clouds generated by the model and a held-out test set.

The rest of this paper is organized as follows: Section 2 outlines the necessary background and building blocks for our work and introduces our evaluation metrics. Section 3 introduces our models for latent representations and generation of point clouds. In Section 4, we evaluate all of our models both quantitatively and qualitatively, and analyze their behaviour. Further results and evaluation can be found in the appendix. The code for all our models is publicly available [1].

## 2 BACKGROUND

**Autoencoders.** Autoencoders (AE - inset) are deep architectures that aim to reproduce their input. They are especially useful, when they contain a narrow *bottleneck layer* between input and output. Upon successful training, the bottleneck layer corresponds to a low-dimensional representation, a *code* for the dataset. The Encoder (E) learns to compress a data point $x$ into its latent representation, $z$. The Decoder (D) can then reproduce $x$ from its encoded version $z$.

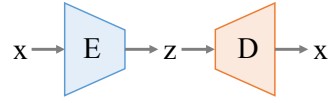

---

[1] http://www.github.com/optas/latent_3d_points

**Generative Adversarial Networks.** GANs are state-of-the-art generative models. The basic architecture (inset) is based on an adversarial game between a *generator* (G) and a *discriminator* (D). The generator aims to synthesize samples that look indistinguishable from real data (drawn from $x \sim p_{\text{data}}$) by passing a randomly drawn sample $z \sim p_z$ through the generator function $G$. The discriminator tries to tell synthesized from real samples. The most commonly used losses for the discriminator and generator networks are:

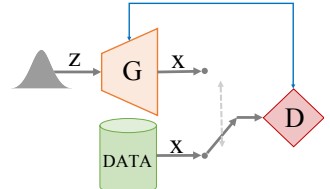

$$J^{(D)}(\boldsymbol{\theta}^{(D)}, \boldsymbol{\theta}^{(G)}) = -\mathbb{E}_{\boldsymbol{x} \sim p_{\text{data}}} \log D(\boldsymbol{x}) - \mathbb{E}_{\boldsymbol{z} \sim p_z} \log\left(1 - D\left(G(\boldsymbol{z})\right)\right), \qquad (1)$$

$$J^{(G)}(\boldsymbol{\theta}^{(D)}, \boldsymbol{\theta}^{(G)}) = -\mathbb{E}_{\boldsymbol{z} \sim p_z} \log D(G(\boldsymbol{z})), \qquad (2)$$

where $\boldsymbol{\theta}^{(D)}, \boldsymbol{\theta}^{(G)}$ are the parameters for the discriminator and the generator network respectively. In addition to the classical GAN formulation, we also use the improved Wasserstein GAN (Gulrajani et al., 2017), which has shown improved stability during training.

**Challenges specific to point cloud geometry.** Point clouds as an input modality present a unique set of challenges when building a network architecture. As an example, the convolution operator – now ubiquitous in image-processing pipelines – requires the signal (in our case, geometry) to be defined on top of an underlying grid-like structure. Such a structure is not available in raw point clouds, which renders them significantly more difficult to encode than e.g. images or voxel grids. Recent classification work on point clouds (PointNet – Qi et al. (2016a)) bypasses this issue by circumventing 2D convolutions. Another issue with point clouds as a representation is that they are unordered - any permutation of a point set still describes the same shape. This complicates comparisons between two point sets, typically needed to define a loss function. This unorderedness of point clouds also creates the need for making the encoded feature permutation invariant.

**Point-set distances.** Two permutation-invariant metrics for comparing unordered point sets have been proposed in the literature (Fan et al., 2016). On the one hand, the *Earth Mover's* distance (EMD) (Rubner et al., 2000) is the solution of a transportation problem which attempts to transform one set to the other. For two equally sized subsets $S_1 \subseteq R^3, S_2 \subseteq R^3$, their EMD is defined by $d_{EMD}(S_1, S_2) = \min_{\phi: S_1 \to S_2} \sum_{x \in S_1} \|x - \phi(x)\|_2$ where $\phi$ is a bijection. Interpreted as a loss, EMD is differentiable almost everywhere. On the other hand, the *Chamfer* (pseudo)-distance (CD) measures the squared distance between each point in one set to its nearest neighbor in the other set: $d_{CH}(S_1, S_2) = \sum_{x \in S_1} \min_{y \in S_2} \|x - y\|_2^2 + \sum_{y \in S_2} \min_{x \in S_1} \|x - y\|_2^2$. It is still differentiable but more computationally efficient.

**Evaluation Metrics for representations and generative models.** In the remainder of the paper, we will frequently need to compare a given set (distribution) of points clouds, whether reconstructed or synthesized, to its ground truth counterpart. For example, one might want to assess the quality of a representation model, in terms of how well it matches the training set or a held-out test set. Such a comparison might be done to evaluate the faithfulness and/or diversity of a generative model, and measure potential mode-collapse. To measure how well a point-cloud distribution $A$ matches a ground truth distribution $G$, we use the following metrics:

*Coverage.* For each point-cloud in $A$ we find its closest neighbor in $G$; closeness can be computed using either CD or EMD, thus yielding two different metrics, COV-CD and COV-EMD. Coverage is measured as the fraction of the point-clouds in $G$ that were matched to point-clouds in $A$. A high coverage score typically indicates that most of $G$ is roughly represented within $A$.

*Minimum Matching Distance (MMD).* Coverage is not representative of the *fidelity* of $A$ with respect to $G$ as matched elements need not be close. To capture fidelity, we match every point cloud of $G$ to the one in $A$ with the minimum distance (MMD) and report the average of distances in the matching. Either of the structural distances can be used, yielding MMD-CD and MMD-EMD. MMD measures the distances in the pairwise matchings, so it correlates with how realistic the elements of $A$ are.

*Jensen-Shannon Divergence (JSD).* The Jensen-Shannon divergence between marginal distributions defined over the euclidean 3D space. Assuming point cloud data that are axis-aligned and a canonical voxel grid in the ambient space; one can measure the degree to which point clouds of $A$ tend to

occupy similar locations as those of $B$. To that end, we count the number of points lying within each voxel across all point clouds of $A$, and correspondingly for $B$ and report the JSD between the obtained empirical distributions.

# 3 REPRESENTATION AND GENERATIVE MODELS

In this section we describe the architectures of our representation and generative models for point clouds, starting from our autoencoder design. Later, we introduce a GAN architecture tailored to point-cloud data, followed by a more efficient pipeline that first learns an AE and the trains a much smaller GAN in the learned latent space, and a simpler generative model based on Gaussian Mixtures.

## 3.1 LEARNING REPRESENTATIONS OF 3D POINT CLOUDS

The input to our AE network is a point cloud with 2048 points ($2048 \times 3$ matrix), representing a 3D shape. The encoder architecture follows the principle of Qi et al. (2016a): 1-D convolutional layers with kernel size 1 and increasing number of features, ending with a "symmetric" function. This approach encodes every point independently and uses a permutation-invariant (symmetric) function to make a joint representation. In our implementation we use 5 1-D conv layers, each followed by a ReLU and a batch-norm layer. The output of the last 1-D conv layer is passed to a feature-wise maximum to produce a $k$-dimensional vector which is the basis for our latent space. The decoder transforms the latent vector with 3 fully connected layers, the first two having ReLUs, to produce a $2048 \times 3$ output. For a permutation invariant objective, we explore both the efficient EMD-distance approximation (Fan et al., 2016) and the Chamfer-Distance as our structural losses; this yields two distinct AE models, referred to as AE-EMD and AE-CD (detailed architecture parameters can be found in Appendix A). To determine an appropriate size for the latent-space, we constructed 8 (otherwise architecturally identical) AEs with bottleneck sizes $k \in \{4, 8 \dots, 512\}$ and trained them with point-clouds of a single object class, under the two losses. We repeated this procedure with pseudo-random weight initializations three times (see appendix, Fig. 15) and found that $k = 128$ had the best generalization error on the test data, while achieving minimal reconstruction error on the train split.

## 3.2 GENERATIVE MODELS FOR POINT CLOUDS

**Raw point cloud GAN (r-GAN).** The first version of our generative model operates directly on the raw $2048 \times 3$ point set input – to the best of our knowledge this work is the first to present a GAN for point clouds. The architecture of the discriminator is identical to the AE (modulo the filter-sizes and the number of neurons), without any batch-norm and with leaky ReLUs (Maas et al., 2013) instead or ReLUs. The output of the last fully connected layer is fed into a sigmoid neuron. The generator takes as input a 128-dimensional noise vector and maps it to a $2048 \times 3$ output by 5 FC-ReLU layers.

**Latent-space GAN (l-GAN).** In our l-GAN, instead of operating on the raw point cloud input, we pass the data through our pre-trained autoencoder, trained separately for each object class with the EMD (or Chamfer) loss function. Both the generator and the discriminator of the GAN then operate on the 128-dimensional bottleneck variable of the AE. Finally,

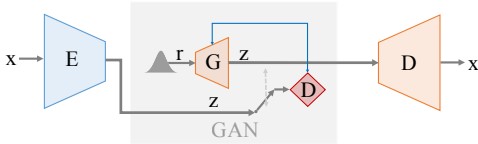

once the GAN training is over, the output of the generator is decoded to a point cloud via the AE decoder. The architecture for the l-GAN is significantly simpler than the one of the r-GAN. We found that very shallow designs for both the generator and discriminator (in our case, one hidden FC layer for the generator and two FC for the discriminator) are sufficient to produce realistic results.

**Gaussian Mixture Model.** In addition to the l-GANs, we also train a family of Gaussian Mixture Models (GMMs) on the latent spaces learned by our AEs. We fitted GMMs with varying numbers of Gaussian components, and experimented with both diagonal and full covariance matrices for the Gaussians. The GMMs can be turned into point-cloud generators by first sampling the latent-space from the GMM distribution and accordingly using the AE's decoder, similarly to the l-GANs.

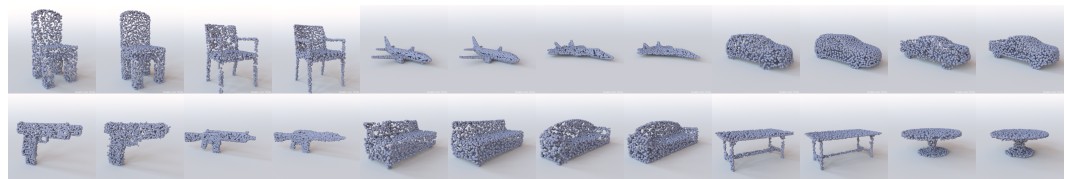

Figure 1: Reconstructions of unseen shapes from the *test* split of the input data. The leftmost image of each pair shows the ground truth shape, the rightmost the shape produced after encoding and decoding using our class-specific AEs.

## 4  EVALUATION AND RESULTS

Our source for shapes is the ShapeNet repository (Chang et al., 2015); we pre-center all shapes into a sphere of diameter 1. Unless otherwise stated, we train specific per-class models, and split the models in each class into training/testing/validation set using a 85%-5%-10% split.

### 4.1  EVALUATING THE LATENT REPRESENTATION

**Classification.**  A common technique for evaluating the quality of unsupervised representation learning algorithms is to apply them as feature extractors on supervised datasets and evaluate the performance of linear models fitted on top of these features. We use this technique to evaluate the performance of the latent "features" computed by our AE. For this experiment to be meaningful, the AE was trained across all different shape categories: we used 57,000 models from ShapeNet from 55 categories of man-made objects. Exclusively for this experiment, we used a bigger bottleneck of 512, increased the number of neurons and applied batch-norm to the decoder as well. To obtain features for an input 3D shape, we feed forward to the network its point-cloud and extract the 512-dimensional bottleneck layer vector. This feature is then processed by a linear classification SVM trained on the de-facto 3D classification benchmark of ModelNet (Wu et al., 2015). Table 1 shows comparative results. Note that previous state of the art (Wu et al., 2016) uses several layers of a GAN to derive a 7168-long feature; our 512-dimensional feature is more intuitive and parsimonious.

| Dataset | SPH[1] | LFD[2] | T-L-Net[3] | VConv-DAE[4] | 3D-GAN[5] | ours - EMD | ours - CD |
|---------|--------|--------|------------|--------------|-----------|------------|-----------|
| MN10 | 79.8% | 79.9% | - | 80.5% | 91.0% | **95.4%** | **95.4%** |
| MN40 | 68.2% | 75.5% | 74.4% | 75.5% | 83.3% | 84.0% | **84.5%** |

Table 1: Classification performance on ModelNet40 and ModelNet10. All methods train a linear SVM with features derived in an unsupervised manner. Comparing to [1] Kazhdan et al. (2003), [2] Chen et al. (2003), [3] Girdhar et al. (2016a), [4] Sharma et al. (2016), [5] Wu et al. (2016).

The decoupling of latent representation from generation allows flexibly choosing the AE loss, which can effect the learned feature. On ModelNet10, which includes primarily larger objects and fewer categories than ModelNet40, the EMD and CD losses perform equivalently. On the other hand, when the variation within the collection increases, CD produces better results. This is perhaps due to its more local and less smooth nature, which allows it to understand rough edges and some high frequency geometric details. Finally, note that since our AEs were not trained on ModelNet, this experiment also demonstrates the domain-robustness of our learned features.

**Qualitative Evaluation.**  To visually assess the quality of the learned representation, we show some reconstruction results in Fig. 1. Here, we use our AEs to encode samples from the *test* split of the ground truth dataset (the leftmost of each pair of images) and then decode them and compare them visually to the input (the rightmost image). These results show the ability of our learned representation to *generalize* to unseen shapes. In addition to reconstruction, our learned latent representation enables a number of interesting shape editing applications, including shape interpolations (Fig. 2), part editing and shape analogies. More results are showcased in Appendix F.

**Generalization Ability.**  Our AEs are able to reconstruct unseen shapes; this is highlighted not only in the results of Figure 1, but also in quantitative measurements of the fidelity and coverage of the

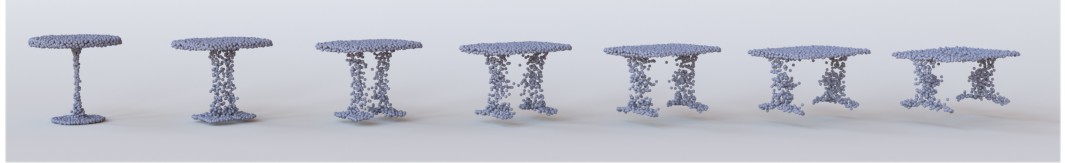

Figure 2: Interpolating between different point clouds, using our latent space representation.

reconstructed ground truth datasets (see appendix, Table 6) and by the comparable reconstruction quality on the training vs. test splits (see appendix, Figure 15).

## 4.2 EVALUATING THE GENERATIVE MODELS

We train and compare a total of five generative models on the data distribution of point-clouds of the *chair* category. We begin by establishing the two AEs with the 128-dimensional bottleneck, trained with the CD or EMD loss respectively – referred to as AE-CD and AE-EMD. Both AEs were stopped at the epoch at which the average reconstruction error with respect to our validation dataset was minimized. We train an l-GAN in each of the AE-CD and AE-EMD latent spaces. In the space associated *only* with the AE-EMD we train a further two models: an identical (architecture-wise) l-GAN that utilizes the Wasserstein objective with gradient-penalty (Gulrajani et al., 2017), and a family of GMMs. Lastly, we also train an r-GAN directly on the point cloud data.

**Model Selection.** All GANs are trained for maximally 2000 epochs; for each GAN, we select one of its training epochs to obtain the "final" model, based on how well the synthetic results match the ground-truth distribution. Specifically, at a given epoch, we use the GAN to generate a set of synthetic point clouds, and measure the distance between this set and the validation set (Section 2). We avoid measuring this distance using MMD-EMD, given the high computational cost of EMD. Instead, we use either the JSD or MMD-CD metrics to compare the synthetic dataset to the validation dataset. To further reduce the computational cost of model selection, we only check every 100 epochs (50 for r-GAN). The epochs at which the various models were selected using the JSD criterion are shown in Table 3. Using the same criterion, we also select the number and covariance type of Gaussian components for the GMM, and obtain the optimal value of 32 components. GMMs performed much better with full (as opposed to diagonal) covariance matrices, suggesting strong correlations between the latent dimensions (see appendix Fig. 17). When using MMD-CD as the selection criterion, we obtain models of similar quality and at similar stopping epochs (see appendix, Table 13); the optimal number of Gaussians in this case was 40.

| Metric | r-GAN | Wu et al. (2016) | l-GAN (AE-CD) | l-GAN (AE-EMD) | l-WGAN (AE-EMD) | GMM (AE-EMD) |
|---|---|---|---|---|---|---|
| JSD | 0.1660 | 0.1705 | 0.0372 | 0.0188 | 0.0077 | **0.0048** |
| Classification | 84.10 | 87.00 | 96.10 | 94.53 | 89.35 | 87.40 |
| MMD-CD | 0.0017 | 0.0042 | 0.0015 | 0.0018 | 0.0015 | **0.0014** |

Table 2: Evaluating 5 generators on **train-split** of *chair* dataset on epochs/models selected via minimal JSD on the validation-split. We also compare against the volumetric approach of Wu et al. (2016). Note that the average classification score attained by the ground-truth point clouds was 84.7%.

**Quantitative Evaluation.** Upon selection of the models, we compare them with respect to their capacity to generate synthetic samples. In two different sets of experiments, we measure how well the distribution of the generated samples resembles both the train and test splits of the ground truth distribution, by using our models to generate a set of synthetic point clouds and employing the metrics from Section 2 to compare against the train or test set distributions respectively. The train split results are reported in Table 2. We also show the average classification probability for those samples being recognized as a chair using the PointNet classifier (Qi et al., 2016a), which is state of the art for classifying point clouds. A similar experiment is ran to measure how well the synthetic samples

| Method | Epoch | JSD | MMD-CD | MMD-EMD | COV-EMD | COV-CD |
|---|---|---|---|---|---|---|
| r-GAN | 1700 | 0.1764 | 0.0020 | 0.1230 | 19.0 | 52.3 |
| l-GAN (AE-CD) | 300 | 0.0486 | 0.0020 | 0.0796 | 32.2 | 59.4 |
| l-GAN (AE-EMD) | 100 | 0.0308 | 0.0023 | 0.0697 | 57.1 | 59.3 |
| l-WGAN (AE-EMD) | 1800 | 0.0227 | 0.0019 | 0.0660 | 66.9 | 67.6 |
| GMM-32-F (AE-EMD) | - | **0.0202** | **0.0018** | **0.0651** | **67.4** | **68.9** |

Table 3: Evaluating 5 generators on **test-split** of *chair* dataset on epochs/models selected via minimal JSD on the validation-split. The reported scores are averages of 3 pseudo-random repetitions. GMM-32-F stands for a GMM with 32 Gaussian components with full covariances.

match the test split dataset; here, we repeat the experiment with three pseudo-random seeds and report the average measurements in Table 3, for various comparison metrics. Perhaps surprisingly, training a simple Gaussian mixture model in the latent space of the EMD-based AE yields the best results in terms of both fidelity and coverage. Furthermore, GMMs are particularly easy to train. Additionally, the achieved fidelity and coverage are very close to the reconstruction baseline, namely, the *lower bounds* for the JSD and MMD achieved by the AE on which the generative models operate (see appendix, Table 6). For example, the AE-EMD achieved an MMD-EMD of 0.05 with respect to the ground truth training data , which is comparable with the MMD-EMD value of 0.06 achieved by the GMMs with respect to the test data. Finally, by comparing Table 2 and Table 3 we can again establish the generalization ability of our models, since their performance for the training vs. testing splits is comparable. This is highlighted in more detail in Fig. 16 in the appendix.

*Note:* The number of synthetic point clouds we generate for the train split experiment is equal to the size of the train dataset. For the test split experiment, as well as for the validation split comparisons done for model selection, we generate synthetic datasets that are three times bigger than the ground truth dataset (the test resp. validation set); this is possible due to the relatively small size of the test resp. validation sets, and helps reduce sampling bias. This is only necessary when measuring MMD or Coverage statistics.

**Fidelity of metrics.**  In Table 3 we note that the MMD-CD distance to the test set appears to be relatively small for the r-GANs. This seeming advantage of the r-GANs is counter to what a qualitative inspection of the results yields. We attribute this effect to the inadequacy of the chamfer distance to distinguish pathological cases. Some examples of such behaviour are showcased in Fig. 3. We show two triplets of images: in each triplet, an r-GAN and an l-GAN is used to generate a synthetic set of point clouds; the left triplet shows an l-GAN on the AE-CD and the right an l-GAN on the AE-EMD. For a given ground truth point cloud from the test set (leftmost image of each triplet), we find its nearest neighbor in each synthetic set under the *chamfer* distance - the middle image in each triplet shows the nearest neighbor in the synthetic results of the r-GAN and the right most image the nearest neighbor in the l-GAN set. We report the distances between these nearest neighbors and the ground truth using both CD and EMD (in-image numbers). Note that the CD values miss the fact that the r-GAN results are visibly of lesser quality. The underlying reason appears to be that r-GANs tend to generate clouds with many points concentrated in the areas that are most likely to be occupied in the underlying shape class (e.g. the seat of chairs in the figure). This implies that one of the two terms in the CD –namely, the one going from the synthetic point cloud to the ground truth– is likely to be very small for r-GAN results. The "blindness" of the CD metric to only *partial* matches between shapes has the additional interesting side-effect that the CD-based coverage metric is consistently bigger than that reported by EMD, as noted in Table 3. Instead, the EMD distance promotes a one-to-one mapping and thus correlates more strongly to visual quality; this means that it heavily penalizes the r-GAN result both in terms of MMD and coverage.

**Training trends.**  We performed extensive measurements during training of our models, to understand their behavior during training, as shown in Fig. 4. On the left, we plot the JSD distance between the ground truth test set and synthetic datasets generated by the various models at various epochs of training. On the right, we also plot the EMD-based MMD and Coverage between the same two sets, where larger marker symbols denote a higher epoch. In general, r-GAN struggles to provide good coverage of the test set no matter the metric used; which alludes to the well-established fact that end-to-end GANs are generally difficult to train. The l-GAN (AE-CD) performs better in terms

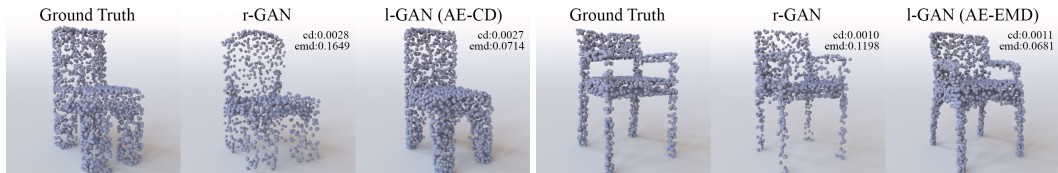

Figure 3: The CD distance is less faithful than EMD to visual quality of synthetic results; in this case it favors r-GAN results, due to the presence of high-density areas in the synthesized point sets.

of fidelity with much fewer epochs as measured by JSD/MMD-EMD, but its coverage remains low. We attribute this to the CD promoting unnatural topologies – cf. Fig. 3 that visually shows this phenomenon. Switching to an EMD-based AE for the representation and otherwise using the same latent GAN architecture (l-GAN, AE-EMD), yields a dramatic improvement in coverage and fidelity. Both l-GANs though suffer from the known issue of mode collapse: Half-way through training, first coverage starts dropping with fidelity still at good levels, which implies that they are overfitting a small subset of the data. Later on, this is followed by a more catastrophic collapse, with coverage dropping as low as 0.5%. Switching to a latent WGAN largely eliminates this collapse, as expected.

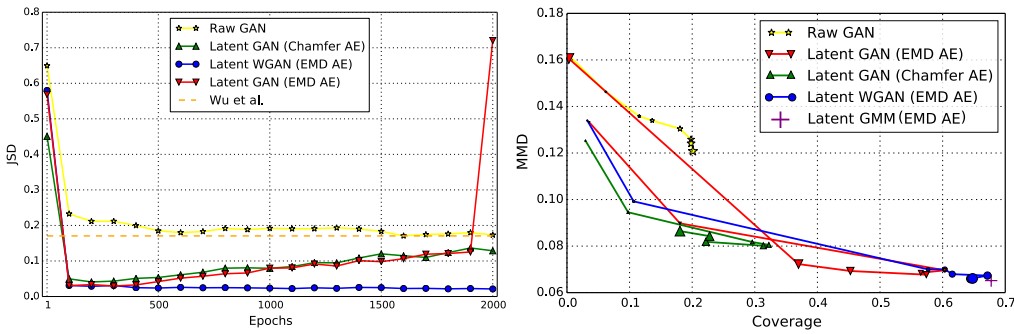

Figure 4: Training trends for the various generative models, in terms of coverage / fidelity to the ground truth **test** dataset. On the right, the curve markers indicate epochs 1, 10, 100, 200, 400, 1000, 1500, 2000, with larger symbols denoting higher epochs. See text for more details.

**Comparisons to voxel-based methods.** To the best of our knowledge we are the first to propose GANs on point-cloud data. To find out how our models fare against other 3D generative methods, in Table 2 and Fig. 4 we compare to a recent voxel-grid based approach (Wu et al., 2016) in terms of the JSD on the training set of the *chair* category - other shape categories can be found in the appendix (Table 10). We convert their voxel grid output into a point-set with 2048 points by performing farthest-point-sampling on the isosurface of the grid values. Per the authors' suggestion, we used an isovalue parameter of 0.1 and isolated the largest connected component from the isosurface. Since Wu et al. (2016) do not use any train/test split, we perform 5 rounds of sampling 1k synthetic results from their models and report the best values of the respective evaluation metrics. The r-GAN mildly outperforms Wu et al. (2016) in terms of its diversity (as measured by JSD/MMD), while also creating realistic-looking results, as shown by the classification score. The l-GANs perform even better, both in terms of classification and diversity, with less training epochs. Note also that the training time for one epoch of the l-GAN is more than an order of magnitude smaller than for the r-GAN, due to its much smaller architecture and dimensionality. For fairness, we acknowledge that since Wu et al. (2016) operates on voxel grids, it is not necessarily on equal standing when it comes to generating point clouds.

**Qualitative evaluation.** In Fig. 5, we show some synthetic results produced by our l-GANs (top row) and the 32-component GMM, both trained on the AE-EMD latent space. We notice high quality results from either model - this highlights the strength of our learned representation, which makes it possible for the simple GMM model to perform well. The shapes (after decoding) corresponding to the 32 means of the Gaussian components can be found in the appendix (Fig. 18), as well as results using the r-GAN (see appendix, Fig. 14). The l-GAN produces crisper and less noisy results than the r-GAN, demonstrating an advantage of using a good structural loss on the decoupled, pre-trained AE.

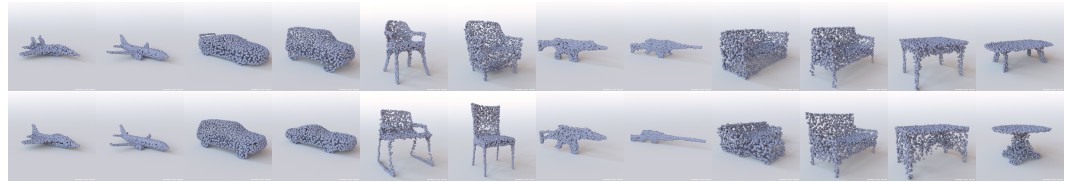

Figure 5: Synthetic point clouds generated by samples produced with l-GAN (top) and 32-component GMM (bottom), both trained on the latent space of an AE using the EMD loss.

**Extensions to multiple classes**    We have performed experiments with an AE-EMD trained on a mixed set containing point clouds from 5 categories (*chair, airplane, car, table, sofa*). The training and testing datasets for this AE were constructed by randomly picking and adding models from each class; 2K models per class for the training set, 200 models for testing and 100 for validation. The multi-class AE has the same bottleneck size of 128 and was trained for 1000 epochs. We compare against the class-specific AEs with the 85-5-10 train-val-test-split, which we trained for 500 epochs. The precise AE model in all cases was selected based on the minimal reconstruction loss on the the respective validation set. On top of all six AEs, we train six l-WGANs for 2K epochs, and evaluate their fidelity/coverage using the MMD-CD between the respective testing sets and a synthesized dataset of 3x the size, as above. It turns out that the l-WGANs based on the multi-class AE perform similarly to the dedicated class-specifically trained ones (Table 4). A qualitative comparison (Fig. 6) also reveals that by using a multi-class AE-EMD we do not sacrifice much in terms of visual quality compared to the dedicated AEs.

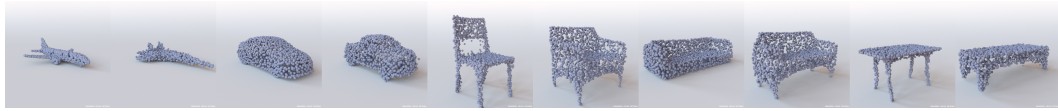

Figure 6: Synthetic point clouds generated by samples produced with l-WGANs trained in the latent space of an AE-EMD trained on a multi-class dataset.

|  | *airplane* | *car* | *chair* | *sofa* | *table* | average | multi-class |
|---|---|---|---|---|---|---|---|
| train | 0.0004 | 0.0006 | 0.0015 | 0.0011 | 0.0013 | **0.0010** | 0.0011 |
| test | 0.0006 | 0.0007 | 0.0019 | 0.0014 | 0.0017 | **0.0013** | 0.0014 |

Table 4: MMD-CD measurements for l-WGANs stopped at the two-thousand epoch and trained on the latent spaces of dedicated (left 5 columns) and multi-class EMD-AEs (right column). The "average" measurement is computed as the weighted average of the per-class values, using the number of train resp. test examples for each class as weights.

**Limitations.**    Fig. 7 shows some failure cases of our models. Chairs with rare geometries (left two images) are sometimes not faithfully decoded. Additionally, the AEs may miss high-frequency geometric details, e.g. a hole in the back of a chair (middle), thus altering the style of the input shape. Finally, the r-GAN often struggles to create realistic-looking shapes (right) for some shape classes – while the r-GAN chairs that are easily visually recognizable, it has a harder time on cars. Designing more robust raw-GANs for point clouds remain an interesting avenue for future work.

## 5    RELATED WORK

A number of recent works (Wu et al. (2016), Wang et al. (2016), Girdhar et al. (2016b), Brock et al. (2016), Maimaitimin et al. (2017), Zhu et al. (2016)) have explored generative and discriminative representations for geometry. They operate on different modalities, typically voxel grids or view-based image projections. To the best of our knowledge, our work is the first to study such representations for point clouds.

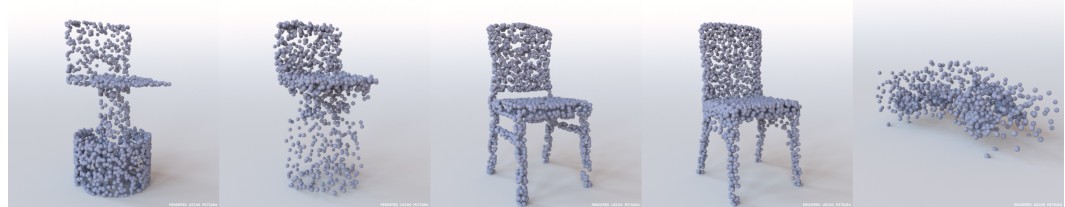

Figure 7: Limitations: The AEs might fail to reconstruct shapes of uncommon/overly detailed geometry (left four images). The r-GAN may synthesize noisy/unrealistic results, cf. a car (right).

Training Gaussian mixture models (GMM) in the latent space of an autoencoder is closely related to VAEs (Kingma & Welling, 2013). One documented issue with VAEs is over-regularization: the regularization term associated with the prior, is often so strong that reconstruction quality suffers (Bowman et al., 2015; Sønderby et al., 2016; Kingma et al., 2016; Dilokthanakul et al., 2016). The literature contains methods that start only with a reconstruction penalty and slowly increase the weight of the regularizer. In our case, we find that fixing the AE before we train our generative models yields good results.

## 6 CONCLUSION

We presented a novel set of architectures for 3D point-cloud representation learning and generation. Our results show good generalization to unseen data and our representations encode meaningful semantics. In particular our generative models are able to produce faithful samples and cover most of the ground truth distribution without memorizing a few examples. Interestingly, we see that the best-performing generative model in our experiments is a GMM trained in the fixed latent space of an AE. While, this might not be a universal result, it suggests that simple classic tools should not be dismissed. A thorough investigation on the conditions under which simple latent GMMs are as powerful as adversarially trained models would be of significant interest.

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

## A   AE Details

The encoding layers of our AEs were implemented as 1D-convolutions with ReLUs, with kernel size of 1 and stride of 1, i.e. treating each 3D point independently. Their decoding layers, were MLPs built with FC-ReLUs. We used Adam (Kingma & Ba, 2014) with initial learning rate of 0.0005, $\beta_1$ of 0.9 and a batch size of 50 to train all AEs.

### A.1   AE used for SVM-based experiments

For the AE mentioned in the beginning of Section 4.1 and which was used for the SVM-related experiments, we used an encoder with $128, 128, 256$ and $512$ filters in each of its layers and a decoder with $1024, 2048, 2048 \times 3$ neurons, respectively. Batch normalization was used between every layer. We also used online data augmentation by applying random rotations along the gravity-(z)-axis to the input point-clouds of each batch. We trained this AE for 1000 epochs with the CD loss and for 1100 with the EMD.

### A.2   All other AEs

For all other AEs, the encoder had $64, 128, 128, 256$ and $k$ filters at each layer, with $k$ being the bottle-neck size. The decoder was comprised by 3 FC-ReLU layers with $256, 256, 2048 \times 3$ neurons each. We trained these AEs for a maximum of 500 epochs when using single class data and 1000 epochs for the sole experiment involving 5 shape classes (end of Section 4.2)

**Remark.** Different AE setups (denoising/regularised) brought no noticeable advantage over our "vanilla" architecture. Adding drop-out layers resulted in worse reconstructions and using batch-norm on the encoder only, sped up training and gave us slightly better generalization error when the AE was trained with single-class data.

## B   r-GAN Details

The discriminator's first 5 layers are 1D-convolutions with stride/kernel of size 1 and $\{64, 128, 256, 256, 512\}$ filters each; interleaved with leaky-ReLU. They are followed by a feature-wise max-pool. The last 2 FC-leaky-ReLU layers have $\{128, 64\}$, neurons each and they lead to single sigmoid neuron. We used 0.2 units of leak.

The generator consists of 5 FC-ReLU layers with $\{64, 128, 512, 1024, 2048 \times 3\}$ neurons each. We trained r-GAN with Adam with an initial learning rate of 0.0001, and $beta_1$ of 0.5 in batches of size 50. The noise vector was drawn by a spherical Gaussian of 128 dimensions with zero mean and 0.2 units of standard deviation.

## C   l-GAN Details

The discriminator consists of 2 FC-ReLU layers with $\{256, 512\}$ neurons each and a final FC layer with a single sigmoid neuron. The generator consists of 2 FC-ReLUs with $\{128, k = 128\}$ neurons each. When used the l-Wasserstein-GAN, we used a gradient penalty regularizer $\lambda = 10$ and trained the critic for 5 iterations per one iteration of the generator. The training parameters (learning rate, batch size) and the generator's noise distribution were the same as those used for the r-GAN.

## D SVM PARAMETERS FOR AUTOENCODER EVALUATION

For the classification experiments of Section 4.1 we used a one-versus-rest linear SVM classifier with an $l_2$ norm penalty and balanced class weights. The exact optimization parameters can be found in Table 5.

| Structural Loss | ModelNet40 | | | ModelNet10 | | |
|---|---|---|---|---|---|---|
| | $C$-penalty | intercept | loss | $C$-penalty | intercept | loss |
| EMD | 0.09 | 0.5 | hinge | 0.02 | 3 | squared-hinge |
| CD | 0.25 | 0.4 | squared-hinge | 0.05 | 0.2 | squared-hinge |

Table 5: Training parameters of SVMs used in each dataset with each structural loss of the AE. *C-penalty*: term controlling the trade-off between the size of the learned margin and the misclassification's rate; *intercept*: extra dimension appended on the input features to center them; loss: svm's optimization *loss* function.

## E AE RECONSTRUCTION QUALITY

Table 6 shows the reconstruction quality of the two AEs (CD- and EMD-based), in terms of the JSD of the reconstructed datasets with respect to their ground truth counterparts. Note that, in general, the quality of reconstruction is comparable between the training and test datasets, indicating that the AEs are indeed able to generalize.

| Method | JSD (Tr) | JSD (Te) | MMD-CD (Tr) | MMD-EMD (Tr) |
|---|---|---|---|---|
| AE-CD | 0.0216 | 0.0243 | **0.0004** | 0.0753 |
| AE-EMD | **0.0028** | **0.0067** | 0.0005 | **0.0527** |

Table 6: Effect of loss-type for AE reconstructions. The EMD loss gives rise to reconstructions with significantly better JSD compared to Chamfer. MMD-measurements favor the AE that was trained with the same loss under which the MMD measurement is computed. (Tr: Train split, Te: Test split)

## F APPLICATIONS OF THE LATENT SPACE REPRESENTATION

For shape editing applications, we use the embedding we learned with the AE-EMD trained *across all* 55 object classes, not separately per-category. This showcases its ability to encode features for different shapes, and enables interesting applications involving different kinds of shapes.

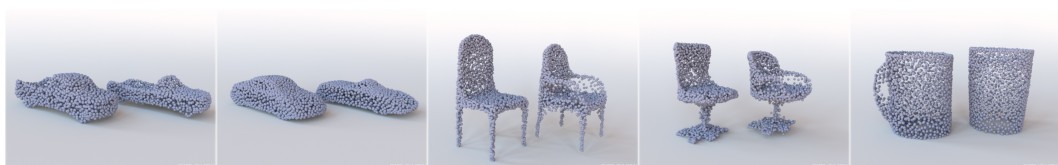

Figure 8: Editing parts in point clouds using vector arithmetic on the AE latent space. Left to right: tuning the appearance of cars towards the shape of convertibles, adding armrests to chairs, removing handle from mug.

**Editing shape parts.** We use the shape annotations of Yi et al.Yi et al. (2016a) as guidance to modify shapes. As an example, assume that a given object category (e.g. chairs) can be further subdivided into two sub-categories $\mathcal{A}$ and $\mathcal{B}$: every object $A \in \mathcal{A}$ possesses a certain structural property (e.g. has armrests, is four-legged, etc.) and objects $B \in \mathcal{B}$ do not. Using our latent representation we can model this structural difference between the two sub-categories by the difference between their average latent representations $\mathbf{x}_\mathcal{B} - \mathbf{x}_\mathcal{A}$, where $\mathbf{x}_\mathcal{A} = \sum_{A \in \mathcal{A}} \mathbf{x}_A$, $\mathbf{x}_\mathcal{B} = \sum_{B \in \mathcal{B}} \mathbf{x}_B$. Then,

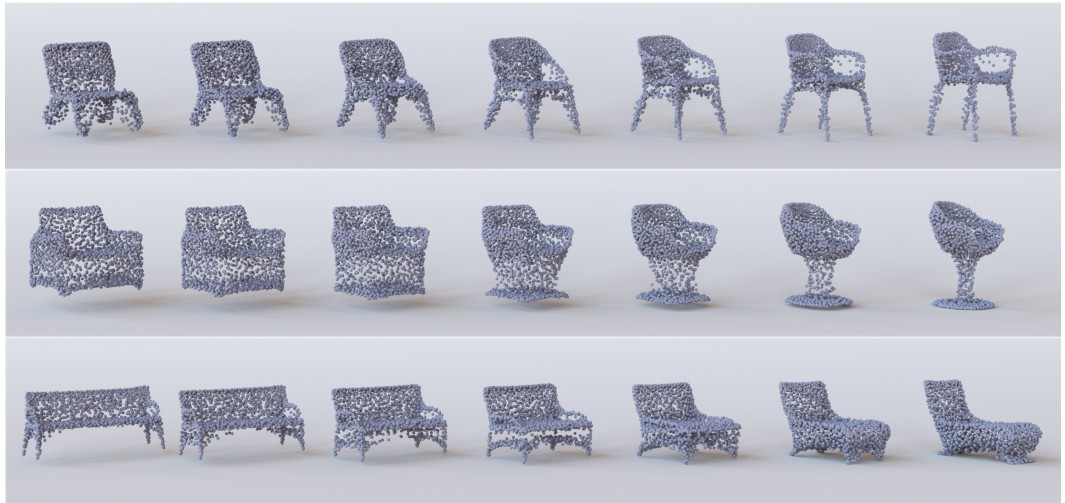

Figure 9: Interpolating between different point clouds, using our latent space representation. Note the interpolation between structurally and topologically different shapes.

given an object $A \in \mathcal{A}$, we can change its property by transforming its latent representation: $x_{A'} = x_A + \mathbf{x}_{\mathcal{B}} - \mathbf{x}_{\mathcal{A}}$, and decode $\mathbf{x}_{\mathcal{A}'}$ to obtain $A' \in \mathcal{B}$. This process is shown in Figure 8. Note that the height of chairs with armrests is on average 13% smaller than the chairs without, which is reflected in the output of this process.

**Interpolating shapes.** By linearly interpolating between the latent representations of two shapes and decoding the result we obtain intermediate variants between the two shapes. This produces a "morph-like" sequence with the two shapes at its end points (Fig. 2, 9). Our latent representation is powerful enough to support removing and merging shape parts, which enables morphing between shapes of significantly different appearance. Our cross-category latent representation enables morphing between shapes of different classes, cfg. the second row for an interpolation between a bench and a sofa.

**Shape analogies.** Another demonstration of the euclidean nature of the latent space is demonstrated by finding "analogous" shapes by a combination of linear manipulations and euclidean nearest-neighbor searching. Concretely, we find the difference vector between $A$ and $A'$, we add it to shape $B$ and search in the latent space for the nearest-neighbor of that result, which yields shape $B'$. We demonstrate the finding in Fig. 10 with images taken from the meshes used to derive the underlying point-clouds to help the visualization. Finding shape analogies has been of interest recently in the geometry processing community Rustamov et al. (2013).

## G    More Comparisons with Voxel-Based Methods

In this section we include preliminary results of point-cloud generators that work in conjunction with voxel-based AEs. We followed the same strategy as we did with the l-GAN but instead of using a point-cloud autoencoder we learned the latent space by an AE that works with occupancy grids of 3D shapes. For generation we used a full-GMM model with 32 centers, which was established as our best model in our previous experiments. We tried two different grid resolutions: $32^3$ and $64^3$ on ShapeNet's chair class. To compare with our established "pure" point-cloud generators we converted the generated voxel-grids into 2048 points by first extracting a mesh from the grids using isosurfacing and then sampling points on the mesh using uniform area-wise sampling. We also compare against Wu et al.'s Wu et al. (2016) voxel-based GANs, which represent the "raw" GAN architecture for the case of voxel grids. For quantitative results and more details see Table 7.

**Discussion.** First, we see that the latent AE-based GMM models outperform Wu et al.'s "raw" GAN architecture by a big margin. In terms of coverage, using the latent representation (voxel GMM)

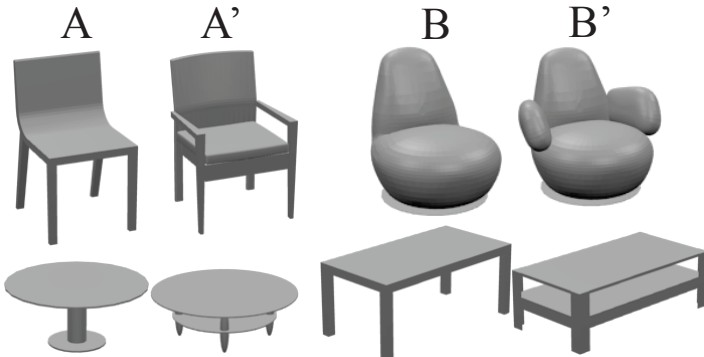

Figure 10: Shape Analogies using our learned representation. Shape $B'$ relates to $B$ in the same way that shape $A'$ relates to $A$.

provides a vast improvement over the "raw" voxel GAN architecture (Wu et al.). This indicates an advantage of using latent representations for generation in the voxel modality as well. Second, we note that the performance of the $64^3$ voxel-based GMM is comparable to the one operating at $32^3$ resolution. This suggests that the main factor affecting fidelity of the results is not the lack of high-frequency details in the ground-truth data. Third, our point-cloud-based models outperform voxel-based models in terms of the fidelity between their output and the ground truth, as measured by the MMD.

The bigger coverage boost of the voxel-based latent-space models compared to the MMD, is likely due to the way the coverage metric is computed: one matches *all* generated shapes against the ground truth, regardless of the quality of the generated shape. Voxel-based models frequently produce shapes with missing components (see Fig. 11); even extremely partial instances ("outliers") will be matched (however poorly) to an arbitrary ground truth model. This effect likely artificially increases the coverage. The histogram in Figure 12 shows distances between the GMM-generated samples and their closest matches in the ground truth. The heavier "tail" for the voxel-based method indicates the presence of such poor quality matchings. Qualitative inspection of the ground truth models that were covered by the voxel-based output but not by the point-cloud output confirmed that the covering came mostly from very poor quality partial shapes.

| Metric | "raw" $64^3$-voxel GAN Wu et al. (2016) | latent $32^3$-voxel GMM-32 | latent $64^3$-voxel GMM-32 | latent $128^3$-octtree GMM-32 | latent point-cloud GMM-32 |
|---|---|---|---|---|---|
| MMD-CD | 0.0046 | 0.0025 | 0.0025 | 0.0024 | **0.0018** |
| MMD-EMD | 0.0915 | 0.0742 | 0.0729 | 0.0750 | **0.0651** |
| COV-CD | 19.6 | 63.5 | 60.3 | 60.9 | **68.9** |
| COV-EMD | 22.4 | 66.6 | 64.8 | 64.7 | **67.4** |

Table 7: MMD and Coverage metrics evaluated on the output of voxel-based methods at resolutions $32^3$ and $64^3$ and (oct-tree based) $128^3$, matched against the chair *test set*, using the same protocol as in Table 3 of the main paper. Our volumetric models use GMMs with full covariances and 32 centers and 64 or 256-dimensional latent codes (for the $32^3$, $64^3$ and $128^3$ respectively). For the mesh conversion we used the marching cubes algorithm ((Lewiner et al., 2003)) with an iso-surface value of $0.5$. The rightmost column shows the results with our point-cloud based GMM.

## G.1 VOXEL AE DETAILS

Our voxel-based AEs are fully-convolutional with the encoders consisting of 3D-Conv-ReLU layers and the decoders of 3D-Conv-Relu-transpose layers. Below, we list the parameters of consecutive layers, listed left-to-right. The layer parameters are denoted in the following manner: (number of filters, filter size). Each conv/conv-tranpose has a stride of 2 except the last layer of the $32^3$ decoder

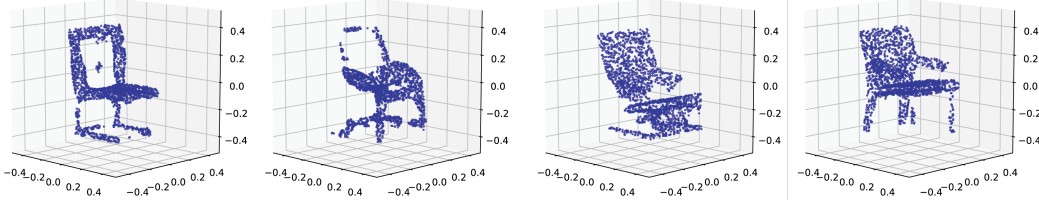

Figure 11: Point clouds extracted from synthetic voxel-based results, after isosurfacing and point sampling. Note the missing components and appearance of noise.

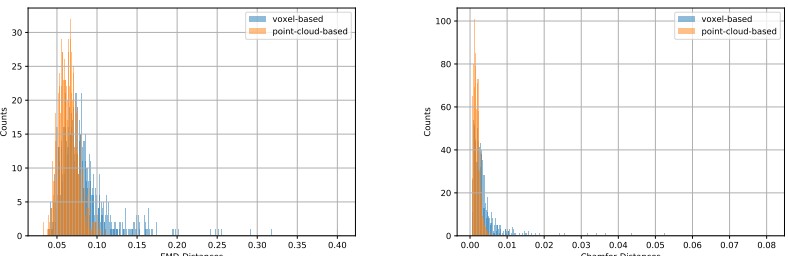

Figure 12: Histograms of MMD-distances: EMD (left) and Chamfer (right), for a purely point-cloud-based generative model (GMM with 32 full-covariance components, in orange) and a voxel-based model (a latent-GAN trained on a voxel-based AE of resolution $64^3$, in blue). Note the larger MMD values for the voxel based approach, indicating results of lower fidelity.

which has 4. In the last layer of the decoders we do not use a non-linearity. The abbreviation "bn" stands for batch-normalization.

- $32^3$ - model
  Encoder: Input $\to (32, 6) \to (32, 6) \to$ bn $\to (64, 4) \to (64, 2) \to$ bn $\to (64, 2)$
  Decoder: $(64, 2) \to (32, 4) \to$ bn $\to (32, 6) \to (1, 8) \to$ Output

- $64^3$ - model
  Encoder: Input $\to (32, 6) \to (32, 6) \to$ bn $\to (64, 4) \to (64, 4) \to$ bn $\to (64, 2)$ $\to (64, 2)$
  Decoder: $(64, 2) \to (32, 4) \to$ bn $\to (32, 6) \to (32, 6) \to$ bn $\to (32, 8) \to (1, 8)$ $\to$ Output

We train each AE for 100 epochs with Adam under the binary cross-entropy loss. The learning rate was 0.001, the $\beta_1$ 0.9 and the batch size 64. To validate our voxel AE architectures, we compared them in terms of reconstruction quality to the state-of-the-art method of Tatarchenko et al. (2017) and obtained comparable results, as demonstrated in Table 8.

| Voxel Resolution | 32 | 64 |
|---|---|---|
| Ours | 92.7 | 88.4 |
| (Tatarchenko et al., 2017) | 93.9 | 90.4 |

Table 8: Reconstruction quality statistics for our dense voxel-based AE and the one of Tatarchenko et al. (2017) for the ShapeNetCars dataset. Both approaches use a 0.5 occupancy threshold and the train-test split of Tatarchenko et al. (2017). Reconstruction quality is measured by measuring the intersection-over-union between the input and synthesized voxel grids, namely the ratio between the volume in the voxel grid that is 1 in both grids divided by the volume that is 1 in at least one grid.

## H    MEMORIZATION BASELINE

Here we compare our GMM-generator against a model that memorizes the training data of the chair class. To do this, we either consider the entire training set or randomly sub-sample it, to create sets of different sizes. We then evaluate our metrics between these "memorized" sets and the point-clouds of test split (see Table 9). The coverage/fidelity obtained by our generative models (last row) is slightly lower than the equivalent in size case (third row) as expected: memorizing the training set produces good coverage/fidelity with respect to the test set when they are both drawn from the same population. This speaks for the validity of our metrics. Naturally, the advantage of using a learned representation lies in learning the structure of the underlying space instead of individual samples, which enables compactly representing the data and generating novel shapes as demonstrated by our interpolations. In particular, note that while some mode collapse is present in our generative results, as indicated by the $\sim 10\%$ drop in coverage, the achieved MMD of our generative models is almost identical to that of the memorization case, indicating excellent fidelity.

| Sample Set Size | COV-CD | MMD-CD | COV-EMD | MMD-EMD |
|---|---|---|---|---|
| Entire \|Train\| | 97.3 | 0.0013 | 98.2 | 0.0545 |
| $1 \times$ \|Test\| | 54.0 | 0.0023 | 51.9 | 0.0699 |
| $3 \times$ \|Test\| | 79.4 | 0.0018 | 78.6 | 0.0633 |
| Full-GMM/32 | | | | |
| $(3 \times$ \|Test\|$)$ | 68.9 | 0.0018 | 67.4 | 0.0651 |

Table 9: Quantitative results of a baseline sampling/memorizing model, for different sizes of sets sampled from the training set and evaluated against the test split. The first three rows show results of a memorizing model, while the third row corresponds to our generative model. The first row shows the results of memorizing the entire training chair dataset. The second and third rows show the averages of three repetitions of the sub-sampling procedure with different random seeds.

## I    MORE COMPARISONS WITH WU ET AL.

In Tables 10, 11, 12 we provide more comparisons with Wu et al. (2015) for the major ShapeNet classes for which the authors have made publicly available their models. In Table 10 we provide JSD-based comparisons for two of our models (see details in main paper 4.2.) In Tables 11, 12 we provide MMD/Coverage comparisons on the *test* split following the same protocol as in Table 3.

| Class | Wu et al. (2016) | L-GAN (AE-EMD) | | Full GMM/32 (AE-EMD) | |
|---|---|---|---|---|---|
| | train+test | train | test | train | test |
| *airplane* | - | 0.0149 | 0.0268 | **0.0065** | 0.0191 |
| *car* | 0.1890 | 0.0081 | 0.0109 | **0.0063** | 0.0108 |
| *rifle* | 0.2012 | 0.0212 | 0.0364 | **0.0092** | 0.0214 |
| *sofa* | 0.1812 | **0.0102** | 0.0102 | **0.0102** | 0.0101 |
| *table* | 0.2472 | 0.0058 | 0.0177 | **0.0035** | 0.0143 |

Table 10: JSD-based comparison between Wu et al. (2016) and our generative models. Full GMM/32 stands for a GM model trained on the latent space of our AE with the EMD structural loss. Note that the l-GAN here uses the same "vanilla" adversarial objective as Wu et al. (2016).

| Class | MMD-EMD | | COV-EMD | |
|---|---|---|---|---|
| | Wu et al. (2016) | Full GMM/32 | Wu et al. (2016) | Full GMM/32 |
| *airplane* | - | 0.0387 | - | 69.6 |
| *car* | 0.0591 | **0.0418** | 28.6 | **65.3** |
| *rifle* | 0.0512 | **0.0459** | 69.0 | **74.8** |
| *sofa* | 0.0773 | **0.0554** | 52.53 | **66.6** |
| *table* | 0.1038 | **0.0615** | 18.35 | **71.1** |

Table 11: EMD based MMD and Coverage comparison between Wu et al. (2016) and our generative model on the *test* split of each class. Full GMM/32 stands for a GM model trained on the latent space of our AE with the EMD structural loss. Note that Wu et al. used *a*ll models of each class for training.

| Class | MMD-CD | | COV-CD | |
|---|---|---|---|---|
| | Wu et al. (2016) | Full GMM/32 | Wu et al. (2016) | Full GMM/32 |
| *airplane* | - | 0.0005 | - | 71.1 |
| *car* | 0.0015 | **0.0007** | 22.9 | **63.0** |
| *rifle* | 0.0008 | **0.0005** | 56.7 | **71.7** |
| *sofa* | 0.0027 | **0.0013** | 42.40 | **75.5** |
| *table* | 0.0058 | **0.0016** | 16.7 | **71.7** |

Table 12: CD based MMD and Coverage comparison between Wu et al. (2016) and our generative model on the *test* split of each class. Full GMM/32 stands for a GM model trained on the latent space of our AE with the EMD structural loss. Note that Wu et al. used *a*ll models of each class for training.

## J   FURTHER EVALUATION AND RESULTS

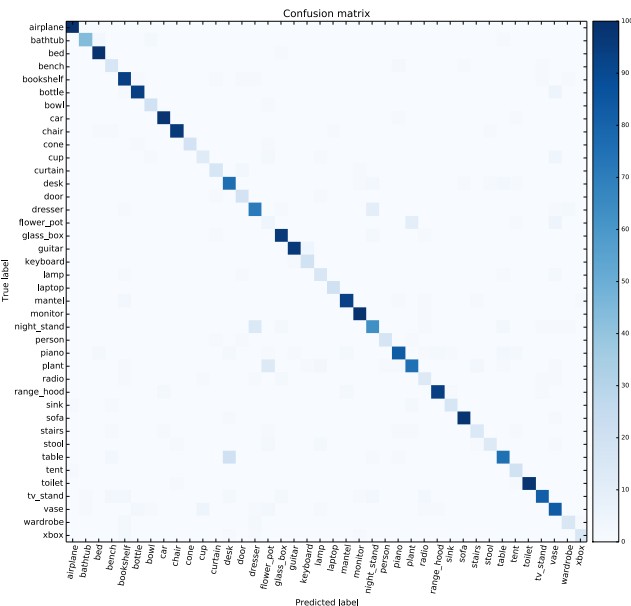

Figure 13: Confusion matrix for the SVM-based classification of Section 4.1, for the Chamfer loss on ModelNet40. The class pairs most confused by the classifier are dresser/nightstand, flower pot/plant. Better viewed in the electronic version.

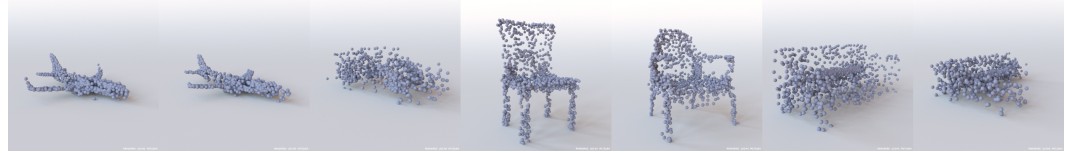

Figure 14: Synthetic results produced by the r-GAN. From left to right: airplanes, car, chairs, sofas.

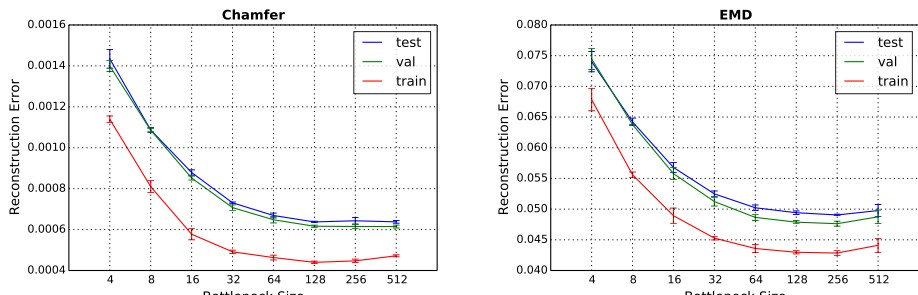

Figure 15: The optimal bottleneck size was fixed at 128 by observing the reconstruction loss of the AEs, shown here for various bottleneck sizes.

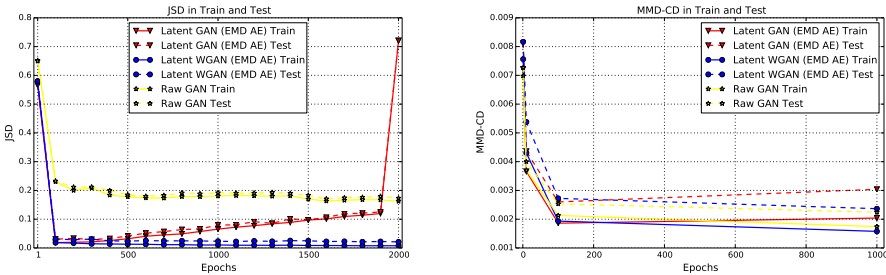

Figure 16: Generalization error of the various GAN models, at various training epochs. Generalization is estimated using the JSD (left) and MMD-CD (right) metrics, which measure closeness of the synthetic results to the training resp. test ground truth distributions. The plots show the measurements of various GANs.

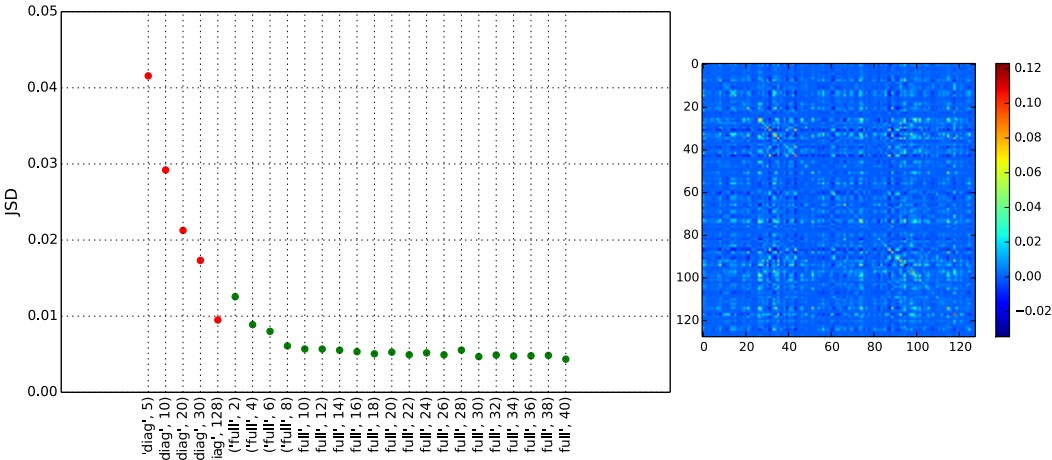

Figure 17: GMM model selection. GMMs with a varying number of Gaussians and covariance type are trained on the latent space learned by and AE trained with EMD and a bottleneck of 128. Models with a full covariance matrix achieve significantly smaller JSD than models trained with diagonal covariance. For those with full covariance, 30 or more clusters seem sufficient to achieve minimal JSD. On the right, the values in a typical covariance matrix of a Gaussian component are shown in pseudocolor - note the strong off-diagonal components.

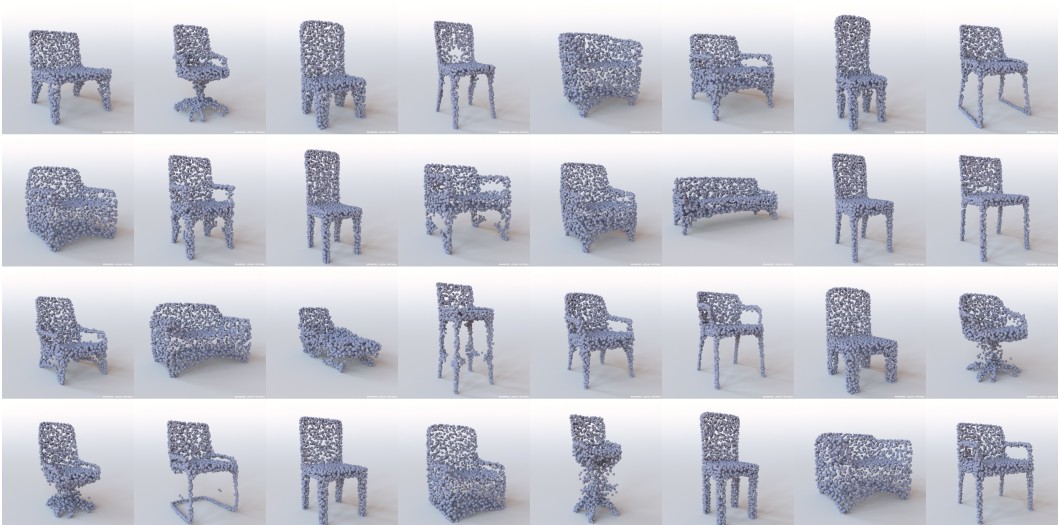

Figure 18: The 32 centers of the GMM fitted to the latent codes, and decoded using the decoder of the AE-EMD.

| Method | Epoch | JSD | MMD-CD | MMD-EMD | COV-EMD | COV-CD |
|---|---|---|---|---|---|---|
| r-GAN | 1350 | 0.1893 | 0.0020 | 0.1265 | 19.4 | 54.7 |
| l-GAN (AE-CD) | 300 | 0.0463 | 0.0020 | 0.0800 | 32.6 | 58.2 |
| l-GAN (AE-EMD) | 200 | 0.0319 | 0.0022 | 0.0684 | 57.6 | 58.7 |
| l-WGAN (AE-EMD) | 1700 | 0.0240 | 0.0020 | 0.0664 | 64.2 | 64.7 |
| GMM-40-F (AE-EMD) | - | **0.0182** | **0.0018** | **0.0646** | **68.6** | **69.3** |

Table 13: Evaluation of five generators on **test-split** of *chair* data on epochs/models that were selected via minimal MMD-CD on the validation-split. The reported scores are averages of three pseudo-random repetitions. Compare this with Table 3. Note that the overall quality of the selected models remains the same, irrespective of the metric used for the selection. GMM-40-F stands for a GMM with 40 Gaussian components with full covariances.

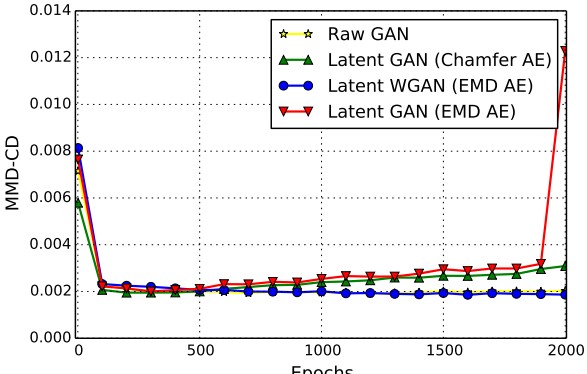

Figure 19: Training trends in terms of the MMD-CD metric for the various GANs. Here, we sample a set of synthetic point-clouds for each model, of size 3x the size of the ground truth test dataset, and measure how well this synthetic dataset matches the ground truth in terms of MMD-CD. This plot complements Fig. 4 (left), where a different evaluation measure was used - note the similar behavior.

