# OpenReview forum: "Learning Representations and Generative Models for 3D Point Clouds"
_ICLR.cc/2018/Conference — Invite to Workshop Track_

### Official Review · AnonReviewer3 · 2017-11-27
**Above average results, but needs more experiments/comparisons**

**Rating:** 6
**Confidence:** 5

**Review:**

This paper introduces a generative approach for 3D point clouds. More specifically, two Generative Adversarial approaches are introduced: Raw point cloud GAN, and Latent-space GAN (r-GAN and l-GAN as referred to in the paper). In addition, a GMM sampling + GAN decoder approach to generation is also among the experimented variations.

The results look convincing for the generation experiments in the paper, both from class-specific (Figure 1) and multi-class generators (Figure 6). The quantitative results also support the visuals.

One question that arises is whether the point cloud approaches to generation is any more valuable compared to voxel-grid based approaches. Especially Octree based approaches [1-below] show very convincing and high-resolution shape generation results, whereas the details seem to be washed out for the point cloud results presented in this paper.

I would like to see comparison experiments with voxel based approaches in the next update for the paper.

[1]
@article{tatarchenko2017octree,
  title={Octree Generating Networks: Efficient Convolutional Architectures for High-resolution 3D Outputs},
  author={Tatarchenko, Maxim and Dosovitskiy, Alexey and Brox, Thomas},
  journal={arXiv preprint arXiv:1703.09438},
  year={2017}
}

In light of the authors' octree updates score is updated. I expect these updates to be reflected in the final version of the paper itself as well.

---

> ### Author Response · Authors · 2017-12-21
> **Addressing Rev. #3's comments**
>
> Thank you for your comments and feedback. We start by pointing out that the primary goal of our work is not to evaluate different 3D modalities. Our stated goal and main focus has been to learn meaningful representations and generative models for point clouds. As a testament to the power of our learned representations, we also reported that they can be used to improve the state of the art in classification (over existing voxel based methods). A full evaluation of different 3D modalities is highly task dependent; we can definitely envision certain tasks where voxels are a good choice. We chose point clouds because we believe that they are relatively unexplored, concise  and natural input modality which appears as the direct output of most 3D range sensing pipelines.
>
> With reference to oct-trees specifically, we agree that the oct-tree-based approach is a powerful one, with the potential to achieve high-resolution results (e.g. Tatarchenko et al.). When occupancy grids are the modality of choice, opting for oct-tree cells as opposed to uniformly sized voxels offers an obvious advantage in terms of the resolution that can be captured within a given memory/space bandwidth. Point clouds are a completely distinct, surface-based representation; it only describes the visible part of a shape, which typically is the only relevant part. As such, compared to volumetric representations (occupancy cells of any kind),  point clouds are typically much more compact.
>
> From a quantitative perspective, the average oct-tree from the Shapenet car category (as per Tatarchenko et al.) requires 389200 bytes to be stored - within this space bandwidth, we could represent the same shape with a point cloud of 32433 points. This would achieve a very high on-surface resolution (15x what is currently shown in our paper), exceeding what 128^3 cells can achieve. We will add such insights in the paper revision and demonstrate visually. Point clouds can be made even more compact by utilizing data-structures such as kd-trees (https://arxiv.org/abs/1704.01222), which could be considered the equivalent, for point clouds, of what oct-trees are for volumetric representations. Exploring this direction for this modality remains an interesting avenue for future work.
>
> That being said, we strongly agree that further exploring the volumetric modality for generation is an interesting direction for study, and we thank you for the pointer. To address this, we have added new comparisons against standard voxel-based methods; we report related observations in a separate message above. Please let us know if these experiments suffice, or of any additional experiments you might have in mind that would better address this question.

---

### Official Review · AnonReviewer2 · 2017-11-27
**Well written paper with new  important results**

**Rating:** 8
**Confidence:** 5

**Review:**

3D data processing is very important topic nowadays, since it has a lot of applications: robotics, AR/VR, etc.

Current approaches to 2D image processing based on Deep Neural Networks provide very accurate results and a wide variety of different architectures for image modelling, generation, classification, retrieval.

The lack of DL architectures for 3D data is due to complexity of representation of 3D data, especially when using 3D point clouds.

Considered paper is one of the first approaches to learn GAN-type generative models.
Using PointNet architecture and latent-space GAN, the authors obtained rather accurate generative model.

The paper is well written, results of experiments are convincing, the authors provided the code on the github, realizing their architectures.

Thus I think that the paper should be published.

---

> ### Author Response · Authors · 2017-12-21
> **Addressing Rev. #2's comments**
>
> Many thanks for your review -- we appreciate your positive feedback. We have since added a number of voxel-based generation experiments and comparisons that you might find interesting - please feel free to refer to the corresponding message above.

---

### Official Review · AnonReviewer1 · 2017-11-27
**interesting approach, but concerns about method**

**Rating:** 5
**Confidence:** 4

**Review:**

Summary:

This paper proposes generative models for point clouds. First, they train an auto-encoder for 3D point clouds,  somewhat similar to PointNet (by Qi et al.). Then, they train generative models over the auto-encoder's latent space, both using a "latent-space GAN" (l-GAN) that outputs latent codes, and a Gaussian Mixture Model. To generate point clouds, they sample a latent code and pass it to the decoder. They also introduce a "raw point cloud GAN" (r-GAN) that, instead of generating a latent code, directly produces a point cloud.

They evaluate the methods on several metrics. First, they show that the autoencoder's latent space is a good representation for classification problems, using the ModelNet dataset. Second, they evaluate the generative model on several metrics (such as Jensen-Shannon Divergence) and study the benefits and drawbacks of these metrics, and suggest that one-to-one mapping metrics such as earth mover's distance are desirable over Chamfer distance. Methods such as the r-GAN score well on the latter by over-representing parts of an object that are likely to be filled.

Pros:

- It is interesting that the latent space models are most successful, including the relatively simple GMM-based model. Is there a reason that these models have not been as successful in other domains?

- The comparison of the evaluation metrics could be useful for future work on evaluating point cloud GANs. Due to the simplicity of the method, this paper could be a useful baseline for future work.

- The part-editing and shape analogies results are interesting, and it would be nice to see these expanded in the main paper.

Cons:

- How does a model that simply memorizes (and randomly samples) the training set compare to the auto-encoder-based models on the proposed metrics? How does the diversity of these two models differ?

- The paper simultaneously proposes methods for generating point clouds, and for evaluating them. The paper could therefore be improved by expanding the section comparing to prior, voxel-based 3D methods, particularly in terms of the diversity of the outputs. Although the performance on automated metrics is encouraging, it is hard to conclude much about under what circumstances one representation or model is better than another.

- The technical approach is not particularly novel. The auto-encoder performs fairly well, but it is just a series of MLP layers that output a Nx3 matrix representing the point cloud, trained to optimize EMD or Chamfer distance. The most successful generative models are based on sampling values in the auto-encoder's latent space using simple models (a two-layer MLP or a GMM).

- While it is interesting that the latent space models seem to outperform the r-GAN, this may be due to the relatively poor performance of r-GAN than to good performance of the latent space models, and directly training a GAN on point clouds remains an important problem.

- The paper could possibly be clearer by integrating more of the "background" section into later sections. Some of the GAN figures could also benefit from having captions.

Overall, I think that this paper could serve as a useful baseline for generating point clouds, but I am not sure that the contribution is significant enough for acceptance.

---

> ### Author Response · Authors · 2017-12-21
> **Addressing Rev. #1's comments**
>
> Thank you for your comments and suggestions. We will incorporate your exposition/text restructuring suggestions in the next revision - below we address the comments pertaining to the technical part of the paper.
>
> A) Evaluation of metrics on models that memorize/randomly sample
> To answer this, we randomly sampled the training set, creating sample sets of 3 different sizes, and evaluated our metrics between these ”memorized” sets and our test set; see https://www.dropbox.com/s/gouvyw1vccqxdkb/memorization-table.png?dl=0 . The coverage/fidelity obtained by our generative models is slightly lower than the equivalent in size (case (b) ), as expected: memorizing the training set produces good coverage/fidelity with respect to the test set when they are both drawn from the same population. This speaks for the validity of our metrics. Naturally, the advantage of using a learned representation lies in learning the structure of the underlying space instead of individual samples, which enables compactly representing the data and generating novel shapes as demonstrated by our interpolations.
>
> B) Comparison to voxel-based approaches
> Since comparing to voxel-based methods was a shared concern, we performed extensive comparisons, which we report in a separate message above.  Please let us know if these experiments suffice, or of any additional experiments you might have in mind that would better address this question. Please note that fully studying latent representations on the voxel modality remains beyond the scope of our work, since point clouds are a distinct representation from voxel grids, with its own set of merits. More details on this can be found in our reply to Rev. #3.
>
> C) rGAN performance
> Indeed, designing significantly better raw GANs directly on point clouds requires further study - we do not claim to have shown that building a point-cloud rGAN with performance en par with (or better than) an lGAN is infeasible. Nevertheless, the fact that our latent representations lead to powerful generation is an interesting and novel result on its own.
>
> D) Simplicity of network architectures
> While this is true, we do not believe is necessarily constitutes a disadvantage of our networks, especially when considering ease of training and reproducibility. Architectures of similar spirit have been shown to work well with point data in the recent literature (PointNet etc.). Our simple models provide a competitive baseline for point cloud learning that establishes the state of the art.
>
> E) Success of latent-space models (including GMMs) in other domains
> This is very much an open question and a great research problem. We cannot assert that latent space models are the way to achieve state-of-the-art results in other problems; follow-up work that explores when this might be the case would be very interesting.  Arguably a big challenge on generative models currently lies in evaluating quality and diversity of their produced samples. Our fidelity and coverage metrics contribute to this evaluation discussion.

---

### Author Response · Authors · 2017-12-21
**Our rebuttal**

We thank all reviewers for their feedback and comments, which we have addressed in the messages below. We look forward to any additional suggestions. Pending reviewers’ approval, we would incorporate all changes below into the appendix of a the next paper revision.

---

### Author Response · Authors · 2018-01-05
**Revision Uploaded**

Dear reviewers,

In the uploaded revision we have incorporated your suggestions and did our best to address your concerns. In the main paper we improved the syntax/language in a handful places and added some missing citations.

Important additions occurred only in the supplementary section; at your suggestion, we will incorporate any of them in the main paper.

Concretely, in the supplementary:

	1.	We added extensive details of our training and architecture parameters to facilitate reproducibility.

	2.	We included the optimal parameters of our SVMs classifiers along with a confusion matrix. By expanding the search space of the SVM parameters we improved the classification scores in ModelNet10 by .1 and .4  in each structural loss.

	3.	We added more comparisons with Wu et al. [Sec. I] and the random-memorization baseline suggested by reviewer-1 [Sec. H].

	4.	We added a section with a new, voxel-based, comparison study [Sec. G].

We appreciate your feedback; it has been invaluable in improving our work.

---

### Decision · Program_Chairs · 2018-01-29
**ICLR 2018 Conference Acceptance Decision**

**Decision:**

Invite to Workshop Track

**Comment:**

This paper compares autoencoder and GAN-based methods for 3D point cloud representation and generation, as well as new (and welcome) metrics for quantitatively evaluating generative models.  The experiments form a good but still a bit too incomplete exploration of this topic.  More analysis is needed to calibrate the new metrics.  Qualitative analysis would be very helpful here to complement and calibrate the quantitative ones.  The writing also needs improvement for clarity and verbosity.  The author replies and revisions are very helpful, but there is still some way to go on the issues above. Overall, the committee is intersting and recommends this paper for the workshop track.